# Development of Machine Learning Models to Evaluate the Toughness of OPH Alloys

**DOI:** 10.3390/ma14216713

**Published:** 2021-11-08

**Authors:** Omid Khalaj, Moslem Ghobadi, Ehsan Saebnoori, Alireza Zarezadeh, Mohammadreza Shishesaz, Bohuslav Mašek, Ctibor Štadler, Jiří Svoboda

**Affiliations:** 1Faculty of Electrical Engineering, University of West Bohemia, Univerzitní 22, 306 14 Pilsen, Czech Republic; masekb@fel.zcu.cz (B.M.); stadler@fel.zcu.cz (C.Š.); 2Department of Inspection Engineering, Abadan Faculty of Petroleum Engineering, Petroleum University of Technology, Abadan 63187-14317, Iran; ghobadi.put92@gmail.com (M.G.); shishesaz@put.ac.ir (M.S.); 3Advanced Materials Research Center, Department of Materials Engineering, Najafabad Branch, Islamic Azad University, Najafabad 15847-43311, Iran; saebnoori@pmt.iaun.ac.ir; 4Department of Safety Engineering, Abadan Faculty of Petroleum Engineering, Petroleum University of Technology, Abadan 63187-14317, Iran; alireza.zarezadeh97@gmail.com; 5Institute of Physics of Materials, Academy of Sciences of the Czech Republic, Žižkova 22, 616 62 Brno, Czech Republic; svobj@ipm.cz

**Keywords:** Oxide Precipitation-Hardened (OPH) alloys, tensile test, toughness, artificial neural network (ANN), particle swarm optimization, ANFIS, Fe-Al-O

## Abstract

Oxide Precipitation-Hardened (OPH) alloys are a new generation of Oxide Dispersion-Strengthened (ODS) alloys recently developed by the authors. The mechanical properties of this group of alloys are significantly influenced by the chemical composition and appropriate heat treatment (HT). The main steps in producing OPH alloys consist of mechanical alloying (MA) and consolidation, followed by hot rolling. Toughness was obtained from standard tensile test results for different variants of OPH alloy to understand their mechanical properties. Three machine learning techniques were developed using experimental data to simulate different outcomes. The effectivity of the impact of each parameter on the toughness of OPH alloys is discussed. By using the experimental results performed by the authors, the composition of OPH alloys (Al, Mo, Fe, Cr, Ta, Y, and O), HT conditions, and mechanical alloying (MA) were used to train the models as inputs and toughness was set as the output. The results demonstrated that all three models are suitable for predicting the toughness of OPH alloys, and the models fulfilled all the desired requirements. However, several criteria validated the fact that the adaptive neuro-fuzzy inference systems (ANFIS) model results in better conditions and has a better ability to simulate. The mean square error (MSE) for artificial neural networks (ANN), ANFIS, and support vector regression (SVR) models was 459.22, 0.0418, and 651.68 respectively. After performing the sensitivity analysis (SA) an optimized ANFIS model was achieved with a MSE value of 0.003 and demonstrated that HT temperature is the most significant of these parameters, and this acts as a critical rule in training the data sets.

## 1. Introduction

Developing new structural alloys for industrial applications requires a shared effort in the commercial sector and a movement towards a green environment. These efforts will thrive if industries use carbon emission-free, safe, and globally available energy sources. One of the primary challenges for structural materials is improving their mechanical properties, mainly focusing on ultimate tensile strength (UTS), elongation, and toughness [1]. The new generations of Oxide Dispersion-Strengthened (ODS) alloys and Oxide Precipitation-Hardened (OPH) alloys are promising candidates for industrial applications, due to their high strength, corrosion resistance, and toughness [2,3,4,5]. Based on the importance of the oxide nanoparticles, they have been widely studied in terms of their morphology, composition, crystallographic structure, and interface relationships with the matrix [6,7,8]. However, further improvement of ODS alloys’ mechanical properties needs appropriate composition designs, which have become a hot topic for researchers. Y_2_O_3_ is one of the typical oxides usually used to develop ODS as well as OPH alloys. However, its strengthening effect is not ideal due to its coarsening at high temperatures [9,10,11]. To reduce the size of oxide dispersoids and produce stable oxide dispersoids, reactive elements, such as Cr, Ti, and Zr, could be added to the Al-free ODS alloys [12,13]. In order to maximize the temperature capability of superalloys, chrome or iron aluminium-based OPH alloys were developed and produced by the mechanical alloying (MA) of powder materials which then followed by Hot Rolling (HR) and Heat Treatment (HT), see the works in [14,15]. This new concept highlighted the novel idea in the processing of OPH alloys: dissolve a required amount of oxygen in the matrix during MA and let a fine dispersion of oxides precipitate during hot consolidation. This microstructural development is highly dependent upon the initial chemical composition and the whole thermomechanical processing background through all processing operations, which still needs optimization [13,16,17].

Over the last few years, machine learning techniques such as artificial neural networks (ANN) and adaptive neuro-fuzzy inference systems (ANFIS) have been developed to analyse and simulate engineering properties [18]. ANN is a statistical simulator inspired by the structure of the human brain; this technique has garnered considerable attention due to its high capabilities and flexibility of use [19]. Linear and nonlinear relations between inputs and outputs are learned without fully calculating mathematical equations, and many engineering problems are solved by this method [20]. In the ANFIS algorithm, the abilities of ANN with fuzzy logic systems are combined. ANFIS converts the logical statements into mathematical correlations [21]. The relation between input-output pairs in a data set would determine a group of rules produced and best membership functions in ANFIS modeling [22]. Ghobadi et al. [23] used ANN and ANFIS models to predict the corrosion resistance of lanolin coatings, and an optimized condition was determined. Support vector regression (SVR) is a simple and powerful technique that is used for regression purposes. It can find nonlinear relationships between variables with high accuracy [24].

Machine learning techniques were used to simulate and predict the mechanical properties of various alloys [25,26,27,28,29]. It has been emphasized that machine learning models could be a promising means for solving engineering problems and studying significant variables [30,31]. Badmos et al. [32] applied the ANN model to simulate the mechanical behavior of ODS alloy and explored the complex conditions of physical models. In order to predict and describe the mechanical properties of alloys based on chemical composition and thermomechanical influence some researchers have developed statistical models. Khalaj et al. [33] used machine learning techniques to predict the hardness of OPH alloys with high accuracy and investigate the effect of each parameter. The results of this research demonstrated that the hardness of OPH alloys was affected by chemical composition, mechanical alloying and heat treatment.

Considering the previous recent findings by the authors [11,14,15,17,20], after hot consolidation using rolling, the microstructure of the OPH alloy shows a relatively homogenous ultra-fine-grained microstructure with a dispersion of very fine (practically invisible in SEM) nano-oxides during several hours of annealing, static recrystallization completed at approximately 1100–1200 °C which then leads to a coarse-grained microstructure strengthened with a homogeneous dispersion of nano-oxides of about 20 nm [5,11]. In the same way, optimized heat treatment improved the UTS, hardness, and elongation by over 100% compared to the initial state [15]. In order to investigate the sensitivity of the toughness to affected parameters, three machine learning approaches—ANN, ANFIS, and SVR—were used to study the significant parameters on affecting the toughness of OPH alloys and simulate the experimental data set.

## 2. Materials and Methods

The new OPH alloy is based on metal powders using powder metallurgy [34]. The main powders (Fe and Al) and other components (Cr, Mo, Ta, Y) are mechanically alloyed in a vacuum low energy ball mill developed by the authors. After sufficient milling, the MA powder is transferred to a low-alloy rolling container with no contact to the air, evacuated, and sealed by welding. Then, it is consolidated using a hot rolling mill in three steps. As first step, the container is rolled under a temperature of 900 °C and rolled to a thickness of 7.5 mm. In the same way, it rolled to thicknesses of 5 mm and 3.2 mm in the next two steps. All the steps have a rolling speed of 0.2 m/s. Finally, the OPH sheet with approximate thickness of 2.5 mm covered on both sides by a 0.3 mm thick scale from the rolling container is produced in this way.

In the current research, different variants of OPH varying in milling time, rolling temperature, and HT were developed to check the effect of each element on the toughness of final semi-products. All variants were produced in a similar way so that the comparison could be available through they could be compared using machine learning methods.

Standard tensile samples were cut for all the variants of the OPH alloys. A waterjet cutting system was used to cut the pieces in a longitudinal direction (parallel to the rolling direction), and then the samples are ground to a final thickness of 2 mm. The authors manufactured purpose-built clamps to hold the samples on the servo-hydraulic MTS machine. All the tensile tests were carried out with a strain rate of 1 × 10^−3^ s^−1^. Three samples were tested for each state, and the average values of the ultimate tensile strength (UTS) and elongation to failure (A) were statistically calculated. The stress–strain curve area was also calculated as toughness using in-house software developed by the authors. The toughness is measured by calculating the area under the stress–strain curve for the OPH alloys. The toughness is calculated from the tensile graph (Figure 1) and based on the following formula:$$U_{T}=\int_{0}^{\varepsilon_{f}}\sigma d\varepsilon$$
where *ε* is strain, *ε_f_* is the strain upon failure, and *σ* is stress. The following figure shows a schematic of the stress–strain curve of an OPH alloy and the desired surface area.

As explained above, fourteen different OPH alloys were prepared and tested to investigate the toughness as a part of the mechanical properties. Machine learning methods (ML) were used to predict the toughness obtained by tensile testing. Systems with different chemical compositions of OPH alloys (Al, Mo, Fe, Cr, Ta, Y, and O), heat treatment conditions, and mechanical alloying conditions were considered as the model inputs and toughness was set as the output.

## 3. Machine Learning Methods Procedure

Three models (ANN, ANFIS, and SVR) were developed to predict and simulate the toughness of OPH alloys. According to the experimental data set, the models were trained, and the performance of each model was evaluated. The accuracy and implementation of the constructed models were calculated by several mathematical errors such as Mean Square Error (MSE):(1)$$\mathrm{MSE}=\frac{1}{n}\sum_{i=1}^{n}{\left(t-o\right)}^{2}$$

The root mean square error (RMSE) was calculated by Equation. (2).
(2)$$\mathrm{RMSE}=\sqrt{\mathrm{MSE}}$$

Mean Absolute Error (MAE) and the absolute fraction of variance (R^2^) are measured by
(3)$$\mathrm{MAE}=\frac{1}{n}\sum_{i=1}^{n}\left|t-o\right|$$
(4)$$R^{2}=1-\frac{\sum_{i=1}^{n-1}{\left(t-o\right)}^{2}}{\sum_{i=1}^{n-1}{\left(t-m\right)}^{2}}$$

In these relations, *n* is the data numbers in training, *t* is experiment data, and *o* is predicted data. The above statistical errors are objective functions between experimental and simulated data. The value of R^2^ ranges between 0 and 1. If a model results in an R^2^ value near 1, it means a slight fluctuation between the experimental and predicted data. If a model results in an R^2^ value close to zero, the most significant difference between the experimental data and the constructed model is explained [35].

### 3.1. Artificial Neural Network (ANN)

An ANN system built several biological neural structures, and its simple parallelism helps solve complex problems that could find suitable relationship parameters [36]. The ANN structure comprises three essential layers: input, hidden, and output layer. According to the experimental tests, twelve parameters influence the toughness of OPH alloys. As shown in Figure 1, twelve parameters were selected as input parameters, and the toughness of OPH alloys was set as the output.

In the ANN structure, there are several neurons in each layer to train the model. As shown in Figure 1, the neuron receives the signals from an input, and the neuron weights each input by a specific weight index (w). The sum of the weighted inputs represents the transfer function f (Ʃ wixi) and the bias (b). Each neuron in a layer is connected to the other neurons, and, finally, the input layer is connected to the output layer by nonlinear mapping. The signal data transfer between each neuron is converted by an activation function or a transfer function [37]. The training process has occurred in the hidden layer, and the performance of the ANN model is highly affected by this layer. The number of neurons in each hidden layer and the number of hidden layers have a significant role in the efficiency of an ANN model. Besides, other essential factors for constructing a suitable ANN structure are the transfer function and training algorithm [33]. In this study, one hidden layer is used as a suitable framework for an ANN structure.

The number of neurons in the hidden layer significantly influences the performance and complexity of the ANN model and helps to avoid underfitting or overfitting. Several topologies were built at various neurons in the hidden layer to determine the optimum number of neurons in the hidden layer. The Levenberg–Marquardt backpropagation (LMBP) training algorithm has been used extensively by researchers in the past and is widely regarded as the best training algorithm [33]. LMBP is used to find a suitable number of neurons. A five-neuron hidden layer architecture was chosen because it resulted in lower error values (Figure 2). The hyperbolic tangent sigmoid transfer function and Purlin transfer function were used for the output in the hidden layer. In this research, we used the ANN toolbox in MATLAB software. Before building an ANN structure, the data set was ranged in a normalized range. According to the data set, 103 pieces of data were collected from experimental tests and were used to construct the models.

In order to train the data set using the ANN model, the input and output values are scaled within the normalized range (before presenting the data). The normalization method improves the ability of simulation and accuracy of the training process. Therefore, all of the data values were set between 0.0 to 1.0. The normalized values (Xnorm) can be calculated by the following equation [33]:(5)$$\mathrm{Xnorm}=\frac{\left(X-\mathrm{Xmin}\right)}{\left(\mathrm{Xmax}-\mathrm{Xmin}\right)}$$
where X is the actual value, Xmin is the minimum value and Xmax is the maximum value of the data set. Table 1 presents the range of values for the variables in this study.

### 3.2. Adaptive Neuro-Fuzzy Inference Systems (ANIFS)

In ANFIS modeling, artificial neural networks and fuzzy system design are mixed, resulting in a robust predictive model [33]. As shown in Figure 3, the ANFIS model comprises five layers that build the fuzzy structure using the “if-then” method and employs the Takagi–Sugeno fuzzy system [38]. In this study, we used the ANFIS toolbox of MATLAB, in which 80% of the total data set was used for training, and 20% of the entire data set was applied for the testing step. Due to the higher efficiency of subtractive clustering (SC) in creating the ANFIS structure, we used this method to generate a fuzzy model [39]. Typical parameters in the SC method are range from influence (RI) and squash factor (SF), which are usually manually changed to build a suitable ANFIS model with an optimal structure [40]. The RI factor ranges from 0.1 to 1 and SF changes from 1 to 7 to find the best ANFIS structure. The number of membership functions is a function of the sub-clustering parameter values, and the number of input membership functions (MF) represents the number of rules. The input and output MF were set as gauss and linear for all structures.

### 3.3. Support Vector Regression (SVR)

The regression method is an efficient method to find the best relationship between the dependent and independent variables. The polynomial regression model may generate in different forms such as vector of random errors, response vector, parameter vector, and design matrix. Nonlinear correlations are found between each variable in polynomial regression models [41]. The support vector machine (SVM) algorithm is one of the most robust and suitable methods for simulating linear and nonlinear data sets [42]. In this research, the linear regression toolbox in MATLAB with the SVM algorithm was developed. As discussed earlier, we used twelve variables as input and toughness as an output to simulate the experimental data set.

## 4. Results and Discussion

### 4.1. The Analysis of the Constructed Models

As discussed in the modelling section, to find the suitable structure of the ANN model, five neurons in a hidden layer with the same transfer function were selected for training the data set. Table 2 shows the comparison between the ANN models with different training algorithms. The results demonstrated that the ANN-4 model had the lowest MSE, MAE (459.22, 15.75), and a higher coefficient of determination than the other models (0.86). Table 3 shows the modeling results for ANFIS constructed models. The higher the RI values, the higher the error; thus, the lower RI value is more suitable for the ANFIS model. Similarly, as the SF factor’s value rises, the performance of ANFIS decreases the optimal value of the SF factor which was 1. As a result, ANFIS-SC8 provided a more suitable arrangement with the lowest RMSE value (RMSE = 0.20).

The optimum ANN structure with five neurons in a hidden layer was selected (e.g., 12-5-1). The LM training algorithm provided a more suitable performance of the ANN model. Consequently, the ANN model for predicting the toughness of OPH alloys has MSE and MAE values of 459.22 and 15.75. Figure 4a shows the correlation between the experimental and predicted values of toughness by the ANN model; the R^2^ value for the ANN model is 0.86, which shows that ANN could find a suitable correlation between the variables.

For ANFIS modeling, several fuzzy clustering structures were developed, and ANFIS-SC8 resulted in higher accuracy. The values of MSE and MAE for the ANFIS model were 0.0418 and 0.0517. The comparison between experimental and simulated toughness values for the ANFIS model is shown in Figure 4b. The ANFIS model predicts the toughness with high accuracy and superior performance (R^2^ = 0.99). The result of the SVR model was presented in Figure 4c, in which acceptable performance was achieved (R^2^ = 0.79). The SVR model could predict the toughness with suitable performance (MSE = 651.68, MAE = 17.42).

### 4.2. Analysis of the Validity and Performance of the Constructed Models

In this section, various criteria were calculated to evaluate the performance and accuracy of each model. As shown in Table 4, several formulae such as R, k, k′, R_o_^2^, and R_o_′^2^ were presented based on previous research [43,44,45,46]. In these relations, *hi* and *ti* represent the observed output and predicted output. Furthermore, the permissible value for the criteria is presented. According to the results, all the models are suitable for predicting the toughness of OPH alloys, and the models fulfilled all the desired performance criteria. Several criteria validated the fact that the ANFIS model results in better conditions and better ability in simulation.

### 4.3. Prediction of the Toughness of OPH Alloys

Ten datasets were collected for the test to compare the models, and these ten datasets were not used to train the models. Moreover, the test datasets were randomly selected to eliminate the problem of the influence of human selection on the results. As shown in Table 5, these ten datasets (e.g., T1 to T10) were reported. Figure 5 and Table 5 illustrate the comparison between the experimental values and the modeling results. It could be understood that the predicted values are compatible with experimental values. However, the ANFIS model exhibited a better and more reliable prediction performance than the ANN and the SVR models. Therefore, we can say that the ANFIS model is more efficient than the other models.

### 4.4. Sensitivity Analysis (SA) of Input Parameters

SA is a suitable technique for investigating the influence of each input on the toughness and for finding the significant input parameters [47]. The inputs were categorized into significant and non-significant parameters. It has been reported that if the non-effective parameters were removed, the accuracy and performance of the model could be enhanced [48]. As discussed in the previous section, the ANFIS model has better performance and accuracy. Therefore, SA was performed on the ANFIS model. The R^2^, MSE, and MAE values for the SA of the ANFIS model are given in Table 6.

The result demonstrates that the ANFIS model is sensitive to the input parameters, especially HT temperature. By removing the HT temperature, the R^2^ decreases considerably from 0.99 to 0.54. Nevertheless, some input variables such as Fe, Mo, and Ta have a more negligible effect on the performance of the ANFIS model (according to Table 6). The HT duration, Rolling Temperature, and Milling time have a higher impact on the model accuracy, and the removal of these parameters causes a higher error of the model.

Furthermore, the reduction of input parameters for Fe, Cr, and Y decreased the MSE from 0.04176 to 0.02615, 0.00171, and 0.06486 respectively. Moreover, the R^2^ value increases and MAE decreases for these input parameters, which showed that removing these parameters could improve the accuracy of the ANFIS model. Therefore, it may be concluded that removing the input parameters results in an increase in MSE and MAE values for the ANFIS model. In particular, reducing HT temperature, HT duration, rolling temperature, milling time, and strain rate increases the error values (see Table 7). Thus, these five parameters are more effective in the performance of the predictive ANFIS model, and the model is more sensitive to these parameters. However, HT temperature is the most significant of these parameters, which acts as a critical rule in training the data sets.

To enable better training of the ANFIS model, it can be optimized by applying the SA technique. Non-effective input parameters, including Fe, Cr, and Y, were removed. In that case, the ANFIS model was developed with nine inputs. Based on that, the predicted data sets versus the actual data obtained from the developed model are shown in Figure 6. It is obvious that the new model with the lowest input parameters achieved better performance with higher accuracy. Above that, by ignoring the non-sensitive input parameters, the overfitting of the model can be avoided and may reduce the complexity and nonlinearity of the data sets. The nature of the input data or configuration directly influences the accuracy of the constructed model [49,50]. By ignoring Fe, Cr, and Y, as non-sensitive parameters, the performance of the ANFIS model can be enhanced. Therefore, a new developed ANFIS model was formed to predict the toughness of OPH alloys with high accuracy.

## 5. Conclusions

Fourteen different OPH alloys were prepared by mechanical alloying from a mixture of powder components, consolidating, and hot rolling. A series of standard tensile tests were performed on different variants of the OPH alloys to investigate the toughness as a part of mechanical properties. Machine learning methods (ML) like ANN, ANFIS, and SVR models were used to predict the toughness obtained by tensile testing. Systems with different chemical compositions of OPH alloys (Al, Mo, Fe, Cr, Ta, Y, and O), heat treatment conditions, and mechanical alloying conditions were considered model inputs and toughness was set as output. The results showed that the proposed strategies can determine the complex behavior of the alloys with an approximate accuracy of 95% and can help the designer predict relevant uncertainties without using analytical calculations. A better understanding of chemical composition to achieve the optimum mechanical properties in a combination of the effectivity of the hybrid model proves the efficiency of the presented models. The value of MSE for ANN, ANFIS, and SVR models was 459.22, 0.0418, and 651.68. Several criteria validated the fact that the ANFIS model results in better conditions and better ability in simulation. Ten datasets were collected for testing the models and it was found that the predicted values are compatible with the experimental values. However, the ANFIS model exhibited a better and more reliable prediction performance. The outcome of SA revealed that the reduction of input parameters for Fe, Cr, and Y decreased the MSE from 0.04176 to 0.02615, 0.00171, and 0.06486, which showed that removing these parameters could improve the accuracy of the ANFIS model. Finally, an optimized ANFIS model was achieved with an MSE value of 0.003.

## Figures and Tables

**Figure 1 materials-14-06713-f001:**
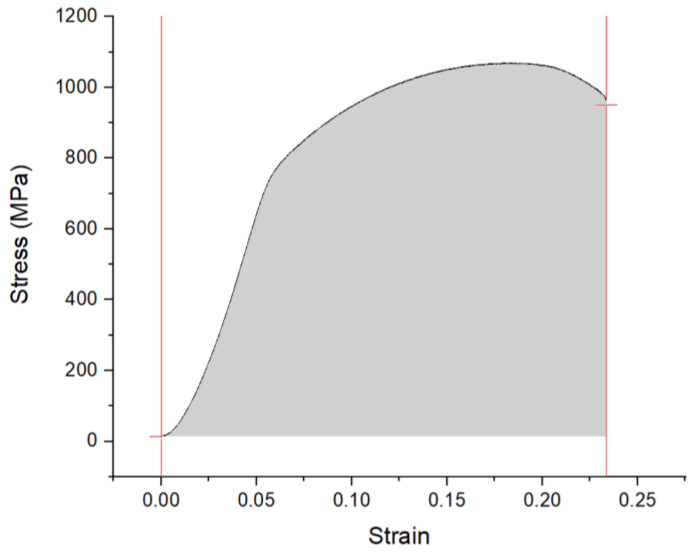
Typical Stress-Strain curve for OPH alloys.

**Figure 2 materials-14-06713-f002:**
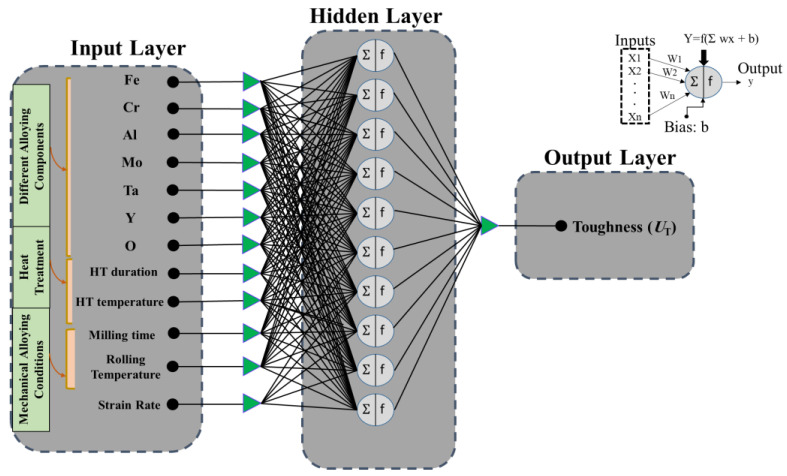
Typical structure for the ANN model used in this study [32].

**Figure 3 materials-14-06713-f003:**
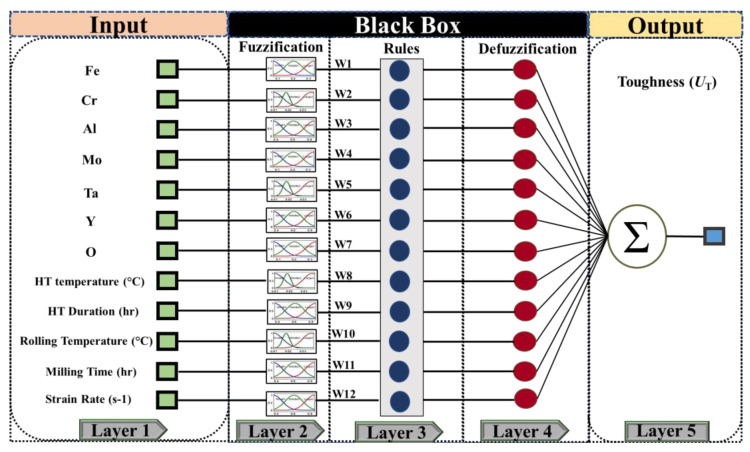
The structure of the ANFIS model for simulation of toughness consisting of 12 inputs and 5 layers [32].

**Figure 4 materials-14-06713-f004:**
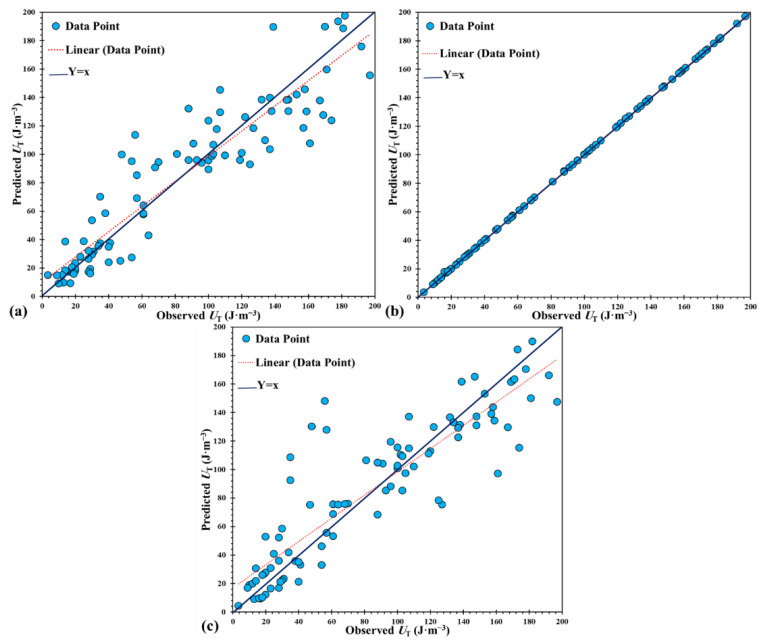
Correlation between the experimental and predicted hardness obtained by the model: (**a**) ANN, (**b**) ANFIS, and (**c**) SVR.

**Figure 5 materials-14-06713-f005:**
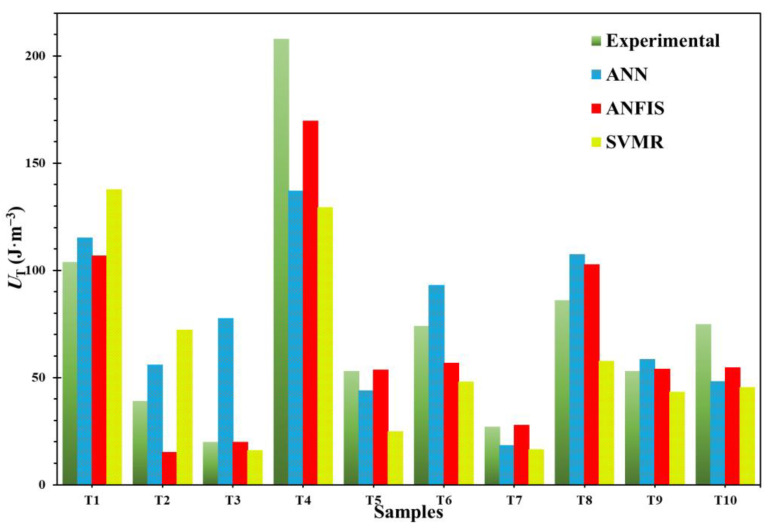
The models for testing data sets predict the value of the hardness of OPH alloy [32].

**Figure 6 materials-14-06713-f006:**
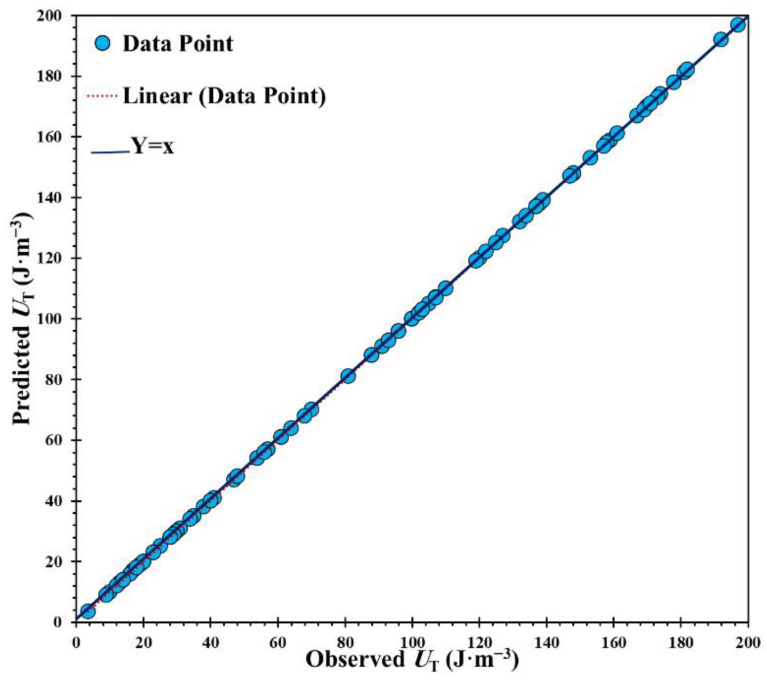
Optimized ANFIS model obtained by removing non-sensitive parameters.

**Table 1 materials-14-06713-t001:** The ranges of parameters in the data set [32].

Variable	Unit	Range
Fe	wt %	0.7106–0.8743
Cr	wt %	0–0.1533
Al	wt %	0.0032−0.1093
Mo	wt %	0–0.0383
Ta	wt %	0–0.0083
Y	wt %	0–0.0364
O	wt %	0–0.0098
Milling time	hours	150–480
Rolling Temperature	°C	850–960
HT duration	hours	0–20
HT temperature	°C	25–1200
Strain Rate	s^−1^	0.001–10
Toughness	J·m^−3^	3.5–208

**Table 2 materials-14-06713-t002:** MSE, MAE, and R^2^ of the five ANN models executed with five neurons in the hidden layer architecture to find the best training algorithm.

ANN Models	Training Algorithm	Symbol	MSE	MAE	R^2^
ANN-1	Resilient backpropagation	RP	776.65	20.85	0.76
ANN-2	BFGS quasi-Newton backpropagation	BFG	785.53	20.37	0.75
ANN-3	Scaled Conjugate Gradient	SCG	812.58	22.65	0.67
*ANN-4*	*Levenberg–Marquardt backpropagation*	*LM*	*459.22*	*15.75*	*0.86*
ANN-5	Conjugate Gradient with Powell/Beale Restarts	CGB	876.25	27.47	0.60

The best model is shown in italics.

**Table 3 materials-14-06713-t003:** The performance and features of alternative ANFIS-SC models.

ANFIS Models	RI	SF	The Number of Input MF	Number of Rules	Epochs	RMSE
ANFIS-SC1	1	2	6	6	20	38.49
ANFIS-SC2	0.9	1.5	11	11	40	262.48
ANFIS-SC3	0.8	3	4	4	60	32.68
ANFIS-SC4	0.7	1.75	14	14	80	29.41
ANFIS-SC5	0.6	2.5	11	11	50	71.36
ANFIS-SC6	0.5	1.25	30	30	30	10.35
ANFIS-SC7	0.4	2.25	26	26	70	14.56
*ANFIS-SC8*	*0.3*	*1*	*61*	*61*	*100*	*0.20*
ANFIS-SC9	0.2	6	26	26	90	56.55
ANFIS-SC10	0.1	7	48	48	15	6.44

The best model is shown in italics.

**Table 4 materials-14-06713-t004:** The validation and performance of each model.

Item	Formula	Condition	ANN	ANFIS	SVR
1	R = $\frac{\sum_{i=1}^{n}\left(hi-\bar{hi}\right)\left(ti-\bar{ti}\right)}{\sqrt{\sum_{i=1}^{n}{\left(hi-\bar{hi}\right)}^{2}}\sqrt{\sum_{i=1}^{n}{\left(ti-\bar{ti}\right)}^{2}}}$	0.8 < R	0.926	0.999	0.892
2	k = $\frac{\sum_{i=1}^{n}\left(hi\times ti\right)}{\sum_{i=1}^{n}hi^{2}}$	0.85 < k < 1.15	0.969	1.0001	0.953
3	k′ = $\frac{\sum_{i=1}^{n}\left(hi\times ti\right)}{\sum_{i=1}^{n}ti^{2}}$	0.85 < k′ < 1.15	0.986	0.999	0.983
4	${R_{o}}^{2}$ = 1 − $\frac{\sum_{i=1}^{n}{\left(ti-hi^{0}\right)}^{2}}{\sum_{i=1}^{n}{\left(ti-\bar{ti}\right)}^{2}}$, $hi^{0}$ = k × *ti*	≈1	0.997	0.999	0.992
5	${R_{o}}^{\prime }{}^{2}$ = 1 − $\frac{\sum_{i=1}^{n}{\left(hi-ti^{0}\right)}^{2}}{\sum_{i=1}^{n}{\left(hi-\bar{hi}\right)}^{2}}$, $ti^{0}$ = k′ × *hi*	≈1	0.999	0.999	0.999

**Table 5 materials-14-06713-t005:** The material parameters of the samples for testing the models with the prediction results.

Material No.	Milling Time (h)	Rolling Temp. (°C)	Annealing	Strain Rate (S^−1^)	Chemical Composition (wt %)	Experimental UT (J·m^−3^)	Predicted UT (J·m^−3^)		
							ANN (R^2^ = 0.60)	ANFIS (R^2^ = 0.88)	SVR (R^2^ = 0.55)
T1	150	925	1200 °C-1 h	0.1	1250 gZ + 90 gAl + 40 gY_2_O_3_ + 50Mo + 12Ta, Z = 83Fe + 17Cr	104	115.2745	106.9551	137.825
T2	230	925	1200 °C-5 h	0.001	1500 gZ + 108 gAl + 70 gY_2_O_3_ + 60Mo + 14Ta, Z = 83Fe + 17Cr	39	55.9562	15.27325	72.43799
T3	230	925	1000 °C-5 h	10	1500 gZ + 108 gAl + 70 gY_2_O_3_ + 60Mo + 14Ta, Z = 83Fe + 17Cr	20	77.6837	19.93451	16.01895
T4	480	960	1100 °C-5 h	0.001	800Fe + 100Al + 30Y_2_O_3_ + 7Y	208	137.1719	169.7749	129.4625
T5	480	960	1000 °C-20 h	0.001	800Fe + 100Al + 15O_2_	53	44.1085	53.7416	24.93518
T6	230	850	1000 °C-20 h	0.001	400 gFe + 80 gCr + 36 gAl + 20 gY_2_O_3_	74	93.2575	56.98798	48.08971
T7	230	865	800 °C-1 h	0.1	1200 gFe + 240 gCr + 108 gAl + 75 gY_2_O_3_	27	18.5657	27.99824	16.53388
T8	230	865	1100 °C-20 h	0.1	1200 gFe + 240 gCr + 108 gAl + 75 gY_2_O_3_	86	107.5968	102.7684	57.76867
T9	230	873	800 °C-5 h	0.1	2400 gFe + 480 gCr + 216 gAl + 120 gY_2_O_3_ + 120Mo	53	58.5657	54.1952	43.41673
T10	230	860	800 °C-1 h	0.001	2400 gFe + 480 gCr + 216 gAl + 120 gY_2_O_3_ + 120Mo	75	48.3064	54.646	45.41388

**Table 6 materials-14-06713-t006:** The R^2^, MSE, and MAE values for the SA of the ANFIS model.

No.	ANFIS Model	R^2^	MSE	MAE
1	11 Input Parameters	0.9999868	0.0417606	0.0517638
2	10 Input Parameters (without Fe)	0.9999917	0.0261573	0.0442478
3	10 Input Parameters (without Cr)	0.9999994	0.0017188	0.0186334
4	10 Input Parameters (without Al)	0.9968289	10.1001732	0.4402436
5	10 Input Parameters (without Mo)	0.9927615	23.0551972	1.4600404
6	10 Input Parameters (without Ta)	0.9505068	157.6411613	2.0653843
7	10 Input Parameters (without Y)	0.9999796	0.0648635	0.1063068
8	10 Input Parameters (without O)	0.9948345	16.4525800	0.5791974
9	10 Input Parameters (without Milling time)	0.9950622	15.7271643	1.1247957
10	10 Input Parameters (without Rolling Temperature)	0.9678562	102.3813008	3.1022166
11	10 Input Parameters (without HT temperature)	0.5434395	1454.195369	27.4952115
12	10 Input Parameters (without HT duration)	0.8766828	392.7787824	13.4959577
13	10 Input Parameters (without Strain rate)	0.9810307	60.4190996	3.2876585

**Table 7 materials-14-06713-t007:** The comparison between the ANFIS model and the Optimized ANFIS model.

	Error	R^2^	MSE	MAE
Model				
ANFIS		0.9999868	0.0417606	0.0517638
Optimized ANFIS		0.9999988	0.0039412	0.0299045

## Data Availability

Not applicable.

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
