# Peer review of "Development of Machine Learning Models to Evaluate the Toughness of OPH Alloys"

_materials, 2021, doi:10.3390/ma14216713_

Round 1
Reviewer 1 Report
The work is quite interesting, but it does not specify clear goals. The article doesn`t have detailed description of the network architecture development. Also, there isn`t indicated reasons that prompted the article authors to develop the models and how the proposed models helped solve their problems. In the hereafter, I inserted some comments.
- It is necessary to decipher the abbreviation ODS and OPH in the Introduction section.
- Clearly state the work goals in the Introduction section.
- In the Introduction section, it is necessary to make a more detailed review of constructing the ANN and ANFIS models for describing and predicting properties that are depend on external thermomechanical effects and the chemical composition. This will make it possible to better show the scientific novelty of the work and formulate the tasks of the work more clearly.
- Apparently, there is a typo in line 122. Is strain rate equal to 10^3 1/ s?
- How many hidden layers were used for the neural network? Based on Figure 1, there were 2 layers, and based on the article text, 1 layer was used.
- How was the workpiece temperature maintained at the strain rate of 0.001 1/s? If there is no way, then we cannot talk about the rolling temperature, but we can talk only about the temperature of the start of rolling.
- It is necessary to describe the neural network training procedure more detail. What was the value of the input neuron set for Cr, Mo, Ta, Y, O and “HT duration” in the case of their zero value (see Table 1)?
Author Response
Dear Reviewer,
Thanks for the valuable comments. We tried our best to fulfil all the requested corrections as well as revise the manuscript. Please see the following answers:
- It is necessary to decipher the abbreviation ODS and OPH in the Introduction section.
Ans: Thanks. It changed in the revised manuscript.
- Clearly state the work goals in the Introduction section.
Ans: Thanks. The goals stated in the last paragraph of the introduction in the revised manuscript.
- In the Introduction section, it is necessary to make a more detailed review of constructing the ANN and ANFIS models for describing and predicting properties that are depend on external thermomechanical effects and the chemical composition. This will make it possible to better show the scientific novelty of the work and formulate the tasks of the work more clearly.
Ans: I appreciate your comment, more details were added to manuscript text in the introduction section.
- Apparently, there is a typo in line 122. Is strain rate equal to 10^3 1/ s?
Ans: Thanks. It was a typo mistake which corrected in the revised manuscript.
- How many hidden layers were used for the neural network? Based on Figure 1, there were 2 layers, and based on the article text, 1 layer was used.
Ans: Thanks for your advice, Figure 1 has been changed and revised. In this study, we used one hidden layer (as mentioned in the manuscript text).
- How was the workpiece temperature maintained at the strain rate of 0.001 1/s? If there is no way, then we cannot talk about the rolling temperature, but we can talk only about the temperature of the start of rolling.
Ans: The rolling machine have heating elements which try to keep the rolling temperature stable like the starting point however the rolling temperature is controlled by a thermos-camera installed on the machine.
- It is necessary to describe the neural network training procedure more detail. What was the value of the input neuron set for Cr, Mo, Ta, Y, O and “HT duration” in the case of their zero value (see Table 1)?
Ans: In order to train the data set by ANN model, it is suitable way to scale the input and output values within the normalized rage (before presenting the data). The normalization method improves the ability of simulation and accuracy of the training process. All of the data values were set between 0.0 to 1.0. So, all of inputs and output parameters were arranged in the specific range and there is no difference between Cr, Mo, Ta, Y, O and “HT duration” with other parameters. Because, they are sorted in normalized values. We added more details of training procedure to the manuscript text and has been highlighted. Thanks for your helpful comment.
Reviewer 2 Report
In this paper, the authors have proposed the best-fitting machine learning model to establish the multivariate correlation between alloy chemistry, heat treatment conditions, and rolling parameters.
- What is the engineering relevance of the parameter “material toughness” measured via the area under the stress-strain curve? How is the material toughness useful for understanding the mechanical performance of OPH alloys?
- Would the notch-toughness be a better parameter to quantify the fracture resistance of OPH alloys?
- The tensile stress-strain curves of the studied OPH alloys may be included in the manuscript to demonstrate their tensile behaviour.
- Which combination of alloying elements, heat treatment conditions, and rolling parameters will produce a better OPH alloy variant?
Author Response
Dear Reviewer,
Thanks for the valuable comments. We tried our best to fulfil all requested comments as well as revise the manuscript. Please see the following descriptions:
- What is the engineering relevance of the parameter “material toughness” measured via the area under the stress-strain curve? How is the material toughness useful for understanding the mechanical performance of OPH alloys?
Ans: In materials engineering, toughness is the ability of a material to absorb energy and perform plastic deformation before fracturing. One definition of toughness is the amount of energy per unit volume that a material can absorb before failure. This toughness measurement differs from the criterion used for fracture toughness, which describes the load-bearing capabilities of defective materials. It is also defined as the resistance of a material to failure during stress. Toughness is related to both strength and ductility. The toughness is associated with the surface below the stress-strain curve. For a material to be tough, it must be both strong and ductile. For example, brittle materials (such as ceramics) that are strong but have limited ductility are not tough. Conversely, very flexible materials with low strength are also not tough. For being tough, a material must withstand both high stresses and strains. In general, strength indicates how much force a material can withstand, while toughness indicates how much energy a material can absorb before it ruptures
- Would the notch-toughness be a better parameter to quantify the fracture resistance of OPH alloys?
Ans: The mechanical properties of a material can be measured and studied in different ways. Each of the parameters related to mechanical properties is applicable somewhere. In this study, the material toughness measured from the surface below the stress-strain curve was investigated which is a general criterion for pre-rupture fracture energy. Because the materials compared from very brittle to soft materials are compared, the fracture energy provides less resolution based on the notched specimen.
- The tensile stress-strain curves of the studied OPH alloys may be included in the manuscript to demonstrate their tensile behaviour.
Ans: The toughness is calculated from the graph and based on the following formula.
where is strain, is the strain upon failure, and is stress. The following figure shows a schematic diagram of the stress-strain curve of an OPH alloy and the desired surface area.
A description as well as typical diagram added to manuscript.
- Which combination of alloying elements, heat treatment conditions, and rolling parameters will produce a better OPH alloy variant?
Ans: The optimize values for input parameters needs to using some algorithms like Genetic algorithm (GA). But, in this study, we developed three machine learning methods to predict the toughness of OPH alloys and investigate the sensitivity of the toughness to affected parameters. So, developing optimized models to find the value of each input parameters could be investigate in future works.
Reviewer 3 Report
From line 309 to line 314, this statement is better to set in a part of Materials and Methods.

Author Response
Dear Reviewer,
Thanks for the valuable comments. We tried our best to fulfil all requested comments as well as revise the manuscript. Please see the following descriptions:
- From line 309 to line 314, this statement is better to set in a part of Materials and Methods.
Ans: A similar paragraph added to “Materials and Method” as recommended by the reviewer.